# Can short PROMs support valid factor-based sub-scores? Example of COMQ-12 in chronic otitis media

Bojana Bukurov[1,2]*, Mark Haggard[3], Helen Spencer[4], Nenad Arsovic[1,2], Sandra Sipetic Grujicic[1,5]

1 Faculty of Medicine, University of Belgrade, Belgrade, Serbia, 2 Clinic for Otorhinolaryngology and Maxillofacial Surgery, University Clinical Centre of Serbia, Belgrade, Serbia, 3 Department of Psychology, University of Cambridge, Cambridge, United Kingdom, 4 Eurotitis Study Group, Cambridge, United Kingdom, 5 Institute for Epidemiology, Belgrade, Serbia

* bojanabukurov@gmail.com, bojana.bukurov@med.bg.ac.rs

## Abstract

### Purpose

Interpretable factor solutions for questionnaire instruments are typically taken as justification for use of factor-based sub-scores. They can indeed articulate content and construct validities of a total and components but do not guarantee criterion validity for clinical application. Our previous documentation of basic psychometric characteristics for a 12-item patient-reported outcome measure in adult chronic otitis media (COMQ-12) justified next appraising criterion validity of sub-scores.

### Methods

On 246 cases at 1st clinic visit, we compared various classes of factor solution, concentrating on the best-fitting 3-factor ones as widely supported. Clinical data offered two independent measures as external criteria: binaural hearing (audiometric thresholds measured via audiometry) for evaluating 'Hearing' sub-score, and oto-microscopic findings for the 'Ear discharge symptoms' sub-score. As criterion for the total, and for semi-generic 'Activities/ healthcare' sub-score, the generic Short Form-36 item set offered a widely used multi-item criterion measure.

### Results

Factor model fit and parsimony again favoured a 3-factor solution for COMQ-12; however insufficient item support and the dominant 1st principal component of variation made sub-scoring problematic. The best solution was bi-factor, from which only the weighted total score met the declared convergent validity standard of r = 0.50. Two of the more specific sub-scores ('Ear discharge symptoms' and 'Hearing') correlated poorly with clinical findings and weighted binaural hearing thresholds.

**Data Availability Statement:** All relevant data are within the manuscript and its Supporting Information files, namely S5 Appendix. If further

inquiries are being made, we will more than happy to answer them.

**Funding:** The authors received no specific funding for this work.

**Competing interests:** The authors have declared that no competing interests exist.

## Conclusion

The COMQ-12 total is acceptably content-valid for general clinical purposes, but the small item set, reflecting excessive pressure for brevity in clinical application, does not well support three criterion-valid factor-based scores. This distinction should be made explicit, and profile sub-scoring discouraged until good convergent and furthermore divergent criterion validities are shown.

## Background

Chronic otitis media (COM) is a bothersome condition with implications for health-related quality of life (HRQoL), mostly due to embarrassing ear discharge and disabling hearing loss [1, 2]. Usually originating in childhood, it is more prevalent in certain sub-populations, and low- and middle-income countries [3]. Surgery remains the treatment of choice in COM patients and is usually directed at eradicating of the disease and/or reducing hearing disability. In active forms of the disease, surgery is considered necessary for preventing or arresting complications [4]. Balanced, informed decision for surgery requires a multi-aspect appraisal to be communicated to the patient for shared understanding and realistic expectations of treatment gains in COM's direct impacts, and in generic HRQoL [5].

The Chronic Otitis Media Questionnaire-12 (COMQ-12) is a short questionnaire covering impacts of COM on HRQoL. It was adapted from items in three questionnaires previously used [6–8]. After the originating publication [9], further information on item scoring, reliability and factor structure came from Phase 1 of the present project [10]. COMQ-12 is largely oriented to direct pathology-linked consequences of COM, so is semi-specific in content. It is self-administered, with 12 questions (7 related to symptoms, 2 to daily activities, 2 to healthcare uptake and 1 overall QoL), each rated on a 6-point numerical scale. The total raw score thus ranges from 0 up to 60 (worst possible HRQoL).

Most published COMQ-12 articles report translations (10 other languages to date) and useful small local-language reference samples [11–19]. Seeking fuller criterion validation might be worthwhile, given factor content validity and the results from two limited validation paradigms: non-nullity for the normal/abnormal ('known-groups') difference as minimal construct validity, and correlations with single global judgements as preliminary face-validation–visual analogue scale (VAS).

The usefulness of all patient-reported outcome measures (PROMs) rests ultimately on the rigor of their validation, clarifying what they actually measure. Despite a multi-centre overview [20], the potential of COMQ-12 to support profiling via its sub-scores has not yet been critically and rigorously addressed. Proceeding to criterion validation for sub-scores pre-supposes their basing in a strongly interpretable and well-fitting factor solution; however even some of the better studies on COMQ-12 have not formally compared relative or absolute goodness-of-fit indices to select most appropriate solutions [21]. The strong 1st principal component (1st PC) of variation for COMQ-12 and the accompanying weak factor structure, had warned early of probably low divergent validities, so of problems for justifiable profiling [9]. For factor model interpretation, we had to clarify also the close relation seen between two of the sub-scores (the 2-item 'Ear discharge symptoms' score and 4-item 'Activities/healthcare'), which fuse in 2-factor solutions [10, 21]. This article therefore formally documents factor solution fits, justifying choice of solution, and proceeds to criterion validation. In this, the expected and confirmed good fit for bi-factor models required that bi-factor, as well as simple CFA, be

considered [22]. More generally, we have followed others' methodological recommendations for attention to long-neglected method issues of power and effect size in otolaryngology, the source discipline [23], also attending to biases and missing data [24] and have further tested metrical assumptions in the item-scoring used.

## Methods

### Participants and data acquired

The Ethical Committee of the School of Medicine, University of Belgrade approved this 2-phase study (decision number 29/II-1), and all patients provided written informed consent. Consecutive adult patients diagnosed with COM (N = 246) were enrolled at a tertiary referral centre over 13 months. Patients filled two questionnaires at each visit, at baseline (1st visit, V1); the role of a 2nd visit (V2) for replicate baseline data on 60 patients [10] and variability reduction is further clarified in the S1 Appendix. Post-operative data at 6 months and 1 year are not used here.

Supplementary demographic data included age, gender, level of education and distance from the capital where the tertiary centre is located. The auditory data available for validating the 'hearing' sub-score included mean air- and bone-conduction thresholds in decibels (Pure Tone Averages, PTAs in dB), for affected and unaffected ears, measured at 0.5, 1, 2 and 4 kHz as recommended [25]. Clinical examination gave details of the disease (onset, duration, form, laterality, previous operations, if any, etc.) before surgery, and the type of operation performed. Oto-microscopic findings underpinned the disease activity assessment (here dichotomised as active/inactive) for validation of the 'Ear discharge symptoms' sub-score.

### Measures used

Previous publications on COM and COMQ-12 had suggested that extracting 3 sub-scores was slightly more justifiable than 2: (one around 'Hearing', one on activities and healthcare uptake, here called 'Activities/healthcare' and one on disease status, here called 'Ear discharge symptoms') [10, 20]. Since COMQ-12 total attempts semi-generic meaning by aggregation and the 'Activities/healthcare' sub-score has semi-generic aspects, we opted for a reputable large generic item set as reliable criterion measure for these two measures. Thus, the second questionnaire used was the translated Serbian version of the widely used HRQoL questionnaire, Short Form-36 (SF-36) [26, 27]. Standard SF-36 scoring modes have several noted deficiencies (see S4 Appendix). To optimise generic nature of an SF-36-based criterion measure item use, we maximised reliability of the adopted criterion measure by using the simplest formulation of a consistency-weighted total, the 1st PC of all SF-36 items. As expected for an item set emphasising physical mobility, the 1st PC form of total score was highly negatively skewed (after scaling, -0.964, with SE 0.155), and was correspondingly non-linearly related to other measures. SF-36 has no natural scale, so to avoid underestimation and false-negative results, we transformed the SF-36 1st PC with inverse natural logarithm (additive constant 1.4 within the bracket); this reduced skewness to near zero (-0.021; SE = 0.155). Inversion also turns positive the typically negative correlations of QoL with disease measures, making expectation for all correlations positive. This transformation also linearised the two hypothesised validity correlations with SF-36 (of COMQ-12 total, and 'Activities/healthcare' respectively, by 0.022 (up to 0.547) and by 0.034 (up to 0.419)).

To maximise metrical precision, and to establish generality [28], we explored three approaches in item scoring (see S2 Appendix) for continuous measurement. For both COMQ-12 and SF-36, the missing-data rate was low; further details on imputation, data-reduction and scoring are given in S2 and S4 Appendices.

### Statistical strategy, power and criterion measures

To restrict multiple testing issues, we hypothesised only four a priori convergent validity correlations.

For COMQ-12 total score and the 'Activities/healthcare' sub-score, we adopted as criterion measure, the $1^{st}$ PC of SF-36 items, and declared r $\geq$0.50 and r$\geq$0.40 a priori as acceptable values respectively. The power scenario was not an a priori sample size calculation, but for r = 0.40, and N = 246, power against nullity (at alpha 0.001, 2-tail) is very high at $(1-\propto)$ = 0.9995, ie almost 100% power. For the 'Ear discharge symptoms' sub-score, the criterion assessment was dichotomised as active/inactive on the independently acquired clinical findings (oto-microscopy) with declared acceptable value $\geq$0.40. For COMQ-12 'Hearing' sub-score, the criterion was the mean binaurally weighted air-conduction hearing threshold from pure-tone audiometry, with declared acceptable value r$\geq$0.50. The binaural weighting is needed to rationally and empirically optimise the differing contributions of the two ears seen in asymmetrical hearing loss [29] (see S3 Appendix).

For the main regular statistical procedures (e.g., correlations, t-tests, GLM (multiple regression), explanatory factor analysis (EFA), and Fisher tests for correlation differences [30], we used SPSS (Version 26.0, SPSS Inc, Chicago, Illinois). The normality of raw descriptives and of model residuals was inspected visually and numerically for skewness and kurtosis as preliminary and in reaction to deficient multivariate normality, as addressed later. Confirmatory factor analyses (CFAs) included bi-factor modelling with the maximum-likelihood estimator in SPSS-AMOS 26, and we followed contemporary standards for reporting and interpreting modelling results with the commonly used goodness-of-fit indices [31–33]. In CFA the 'factors' are technically latent variables, but we retain the 1-word more widely understood term.

## Results

The descriptives in Table 1 show the sample's main clinical and demographic characteristics. The 246 patients were mostly not highly educated, with predominantly longstanding and active disease, and a wide symmetrical age distribution with mean (M) 41.61 years and Standard Deviation (SD) 15.73. The raw total of COMQ-12 items, marking the general sample severity, had a usefully symmetrical distribution, with M 25.41 and SD 11.16. After assessment,

**Table 1. Basic descriptives of sample on main demographic and clinical variables.**

| Descriptor | | N (%) |
|---|---|---|
| **Distance of residence:** | Up to 100 km | 135 (54.9) |
| (**from Belgrade**) | More than 100 km | 91 (37.0) |
| | Missing | 20 (8.1) |
| **Education:** | Primary, Lower secondary, Missing | 129 (52.4) |
| | Upper secondary, post-secondary, 1st stage tertiary | 117 (47.6) |
| **Disease activity stage:** | Inactive | 81 (32.9) |
| | Active Mucosal | 71 (28.9) |
| | Active Squamous | 94 (38.2) |
| **Duration of disease:** | 1–8 years | 99 (40.2) |
| | 8–32 years | 137 (55.7) |
| | Missing | 10 (4.1) |

Of the 246 patients, 47.2% were male; 72.4% had unilateral and 27.6% bilateral disease. Stated duration of disease was initially coded logarithmically: 1–2, 2–4, 4–8 years etc. but for simplicity, was dichotomised (as here) at nearest-to-median boundary.

41 patients were treated conservatively and of the 205 (83%) with surgical treatment recommended, 167 accepted. Hearing thresholds by air conduction showed moderate conductive hearing loss on the (more) affected ear, with mean air conduction (PTA) 54.87 decibels (dB), SD 18.99 and bone conduction 27.18 dB (SD 13.97). For the less affected ear, values were only mildly impaired: air PTA 32.21 dB (SD 18.19) and bone PTA 22.12 (10.91) respectively.

### Preliminary EFA solutions

Unscaled Kaiser-Meyer-Olkin value of 0.767 and Bartlett Chi-sq value of 945.26, df = 66, p<0.001, showed that factor-analysis was justified. The 1st PC again dominated, explaining 34.41% of variance. To avoid a large detailed 3x3 (scoring basis, factor structure) tabulation, we first scoped factor structure issues on 2-Factor (2-F), 3-F and 4-F solutions in exploratory factor analyses (EFA rotated Varimax), based on the raw item scoring (ie numerical rating). This gave respectively solution Rsq values (unscaled) of 0.472, 0.568 and 0.645, with respective rotated last-extracted eigenvalues (LREV) of 2.77, 1.67 and 1.69. As expected, 2-F gave a simple but not very useful solution interpretable as 'Hearing' versus the rest, whilst 4-F separated the 'Activities/healthcare' items, attracting item 6 (dizziness) to the former; this is unsatisfactory, and reflects overfitting in EFA when an entire item set is forced into use. For CFA, we therefore focussed on variants of 3-F as the preferred structure a priori from the literature and on scaled versions of items (details in S2 Appendix).

### Formal comparison of most relevant factor solutions using CFA

The EFA results seeded the CFAs, followed by deletion of links for lowest item loadings (as standardised regression weight—SRW). The minimum retained link SRW in the simple CFA was 0.16, for Q6 on the 'Ear discharge symptoms' factor. Both simple CFA and bi-factor CFA were be considered, because in a dataset with strong 1st PC, bi-factor generally improves both fit and the capture of fine structure. In the 3-F simple CFA model, two cross-loading items, question 6 ('dizziness') and 12 ('ear problems get you down?') were retained with respective links (to 'Hearing' also 'Ear discharge symptoms', and to 'Hearing' also 'Activities/healthcare'), because their deletion degraded the model fit, and so would erode sub-score support for profiling. All three inter-factor correlations (IFCs) were retained in simple CFA, the lowest being r = 0.338, confirming remaining factor interdependence in CFA.

Item contents and SRW loadings in simple CFA for 2-F and 3-F solutions are given in Table 2 (upper two fields), including the retained cross-loadings. As in EFA, the 'extra' factor in 3-F splits the second factor from 2-F into 'Ear discharge symptoms' and 'Activities/healthcare'. The 3-F simple CFA gives fair, not excellent, fit index values (Chi sq = 138.145, DF = 49, RMSEA = 0.086; AIC = 220.145, AIC saturated = 180, delta AIC = 40.145; CFI = 0.906). Qualitatively, this pattern differs little from the previously published simple CFA for Spanish COMQ-12 [21] and it approaches an earlier standard of RMSEA 0.08. Its falling short of the current standard of excellence (expressed as RMSEA < 0.05) is most likely due to the paucity and some inhomogeneity of items [34]. The fit for the simple 2-F CFA solution (Chi sq = 243.147, DF = 51, RMSEA = 0.124, CFI = 0.797; AIC = 321.15, AIC saturated = 180, delta AIC = 141.147) was not acceptable, with parsimony- adjusted delta AIC poorer by about 100.0 than the one for 3-F.

The bi-factor (3-F plus general factor) solution in Fig 1 and the lowest field of Table 2 approached excellent fit (Chi sq = 90.048, DF = 44, RMSEA = 0.065, CFI = 0.951). The bi-factor solution also achieved remarkably high parsimony with AIC 'default' = 182.048, AIC saturated = 180.0 (delta AIC = 2.048). For readers unfamiliar with modelling indices and their properties, this means roughly that the bi-factor fit is about as good as could be reasonably

**Table 2. COMQ-12 item loadings expressed as standardised regression weights (SRWs) for three factor structure models in CFA.**

| Item → | 1 | 2 | 3 | 4 | 5 | 6 | 7 | 8 | 9 | 10 | 11 | 12 |
|---|---|---|---|---|---|---|---|---|---|---|---|---|
| **Solution & factor label ↓** | | | | | | | | | | | | |
| **Simple 2-F** | | | | | | | | | | | | |
| **F1 'Hearing'** | | | 0.830 | 0.882 | 0.538 | 0.244 | 0.500 | | | | | 0.355 |
| **F2 'Activities, healthcare *plus* Ear discharge symptoms'** | 0.483 | 0.419 | | | | 0.222 | | 0.501 | 0.434 | 0.725 | 0.803 | 0.395 |
| **Simple 3-F** | | | | | | | | | | | | |
| **F1 'Hearing'** | | | 0.826 | 0.879 | 0.543 | 0.293 | 0.506 | | | | | 0.395 |
| **F2 'Activities/ healthcare'** | | | | | | | | 0.475 | 0.411 | 0.770 | 0.847 | 0.344 |
| **F3 'Ear discharge symptoms'** | 0.855 | 0.757 | | | | 0.160 | | | | | | |
| **Bifactor 3-F** | | | | | | | | | | | | |
| **General** | 0.390 | 0.292 | 0.611 | 0.681 | 0.642 | 0.477 | 0.629 | 0.346 | | 0.292 | 0.358 | 0.618 |
| **F1 'Hearing'** | | | 0.616 | 0.555 | | -0.085 | | | | | | |
| **F2 'Activities/ healthcare'** | | | | | | | | 0.327 | 0.386 | 0.725 | 0.760 | 0.227 |
| **F3 'Ear discharge symptoms'** | 0.764 | 0.695 | | | | | | | | | | |

Abbreviated keywords for questionnaire items: 1 Ear drainage; 2 Smelly ear; 3 Hearing at home; 4 Hearing in noise; 5 Ear discomfort; 6 Dizziness; 7 Tinnitus; 8 Restricted activities; 9 Unable to wash; 10 General practitioner visits for ear problems; 11 Use of medicines for ear problems; 12 Ear problems 'get you down'. Double-row entries per column represent cross-loading for simple versions of CFA, but for bi-factor analysis the 9 out of 12 dual instances represent dual loadings on the general and one specific factor. The bi-factor general link to item 9 had to be suppressed to enable the model to run, and two weak loadings from any specific factor to Q5 or to Q7 likewise. Of three low loadings (SRW<0.15) considered for dropping for simplicity, 'ear discharge symptoms' to Q6 and 'hearing' to Q12 were then dropped. The third, although low, 'hearing' to Q6, had to be retained to enable the model to run, the low or negative sign making it a contrasting anchor not to be considered part of hearing disability. In the bi-factor model, 'Hearing' and 'Ear discharge symptoms' both become under-sampled, with only 2 strongly loading-items.

expected, given its high parsimony. Therefore, further statements refer to this model and multivariate normality was examined for the preferred solution. As there was a slight kurtotic infringement of multivariate normality (Mardia multivariate kurtosis 6.39, CR 2.74) [35], we bootstrapped the model 1000 times for re-sampled 'conservative', ie distribution-free, p-values. There were only two items with marginal p-values under normality assumptions, in the 'Hearing' factor, and these were also highly kurtotic. Taking the cross-loading item 6 with SRW = 0.085 as reference, we accepted the retention of these links based on their bootstrapped p-values of 0.00039 and 0.00050. The other 10 COMQ-12 items all had loadings (SRW) above 0.35, many above 0.5, with the exception of one general factor link at 0.23. All bootstrapped p-values undercut p = 0.005 for specific links, and 0.025 for general factor links. Thus, multivariate kurtosis is not a concern for the acceptance of model inks. This summary refers to the better-fitting, bi-factor, model but consistent comparisons held for simple CFA.

An IFC or alternatively regression link (SRW = 0.273), between factors 2 and 3 (ie 'Ear discharge symptoms' and 'Activities/healthcare') was necessary to permit convergence of model estimates, recalling the competitiveness of 2-F EFA solutions. However, a 2-F bi-factor model had to be rejected structurally; an illogical sign reversal between the above two factors made the resulting scores unsuitable for interpretation and clinical application.

## Convergent and divergent validation of total and sub-scores

Table 3 summarises as Pearson correlations the criterion validities of COMQ-12 total and sub-scores, according to simple CFA and bi-factor models. Emboldening represents the prediction and requirement for highest correlations in their row, in showing convergent validity. Obtained emboldened correlations are indeed mostly the highest, but they are modest overall, and neither solution produces r>0.4 for 'Ear discharge symptoms'. For defining a total score,

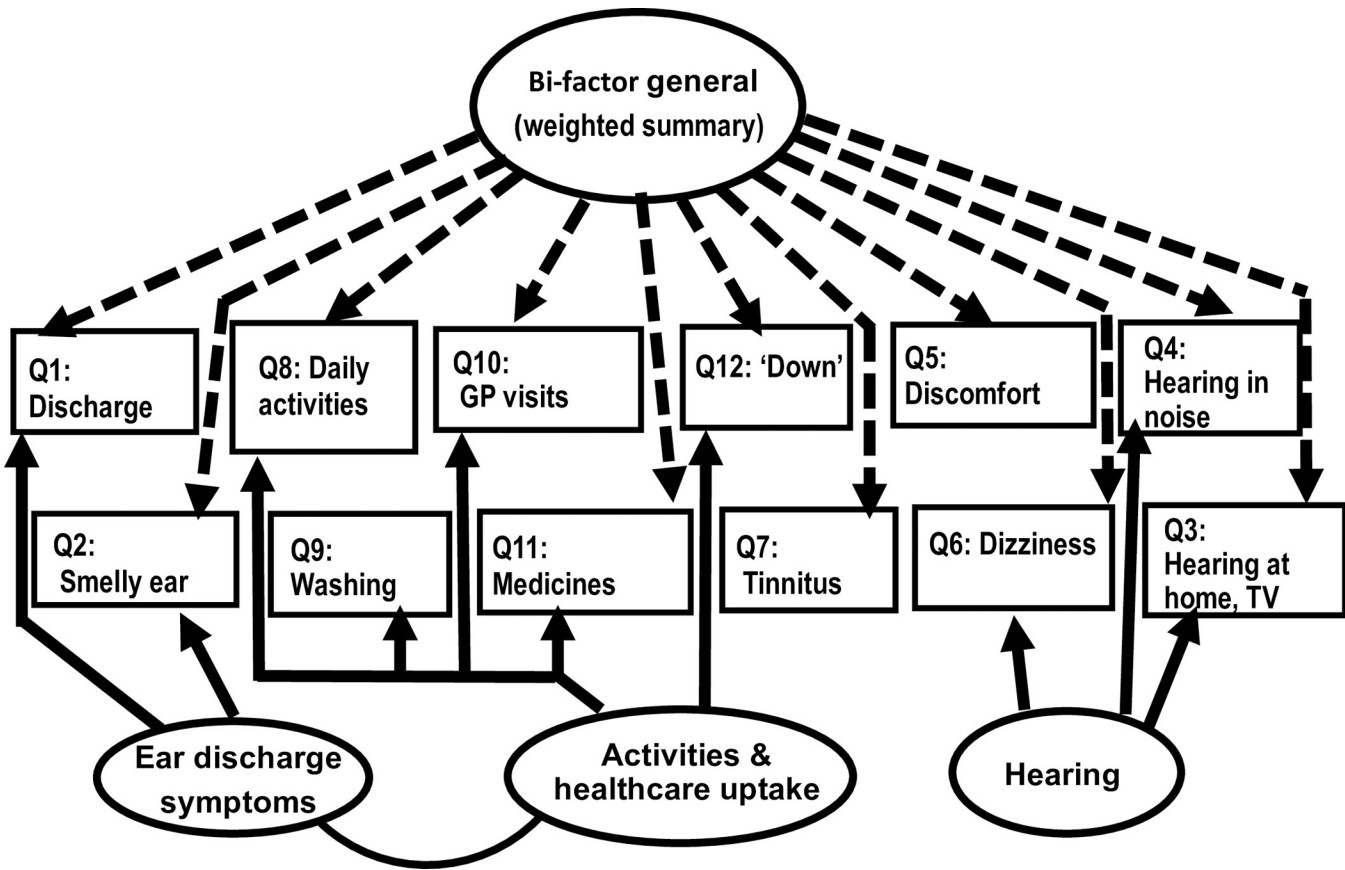

**Fig 1. Simplified graphic of CFA for Bi-factor solution to COMQ-12 items.**

bi-factor versus simple solution makes little difference to the validity correlation values with SF-36; both exceed declared acceptability cut-off for total. The bi-factor solution is necessary to allow correlation of 'Activities/healthcare' with SF-36 to exceed the declared cut-off.

Assessing divergent validities under bi-factor solution demands six tests for correlation difference (See Table 3 footnote for Fisher Z and p-values). Divergent validity is shown when the non-emboldened correlations in the same row are lower than the corresponding emboldened ones. There is no general convention for size of required difference, and PROM development rarely gets this far; we adopted correlation magnitude differences of ≥0.300 as good, and ≥0.200 as fair. For-bi-factor sub-scores, in Column 1 the 0.335 at the bottom left differs from the two other bi-factor rows' entries by > 0.300 (good divergence), and the significance tests in the footnote generalise this example. Simple CFA does not achieve that. Improved divergent validity with bi-factor is expected from its co-extraction of a general factor, central to bi-factor purpose and usefulness. Indeed, under bi-factor solution, the required correlation difference for 'hearing' exceeds 0.200 in one instance of PROM variable not required for convergent validity (vs generic SF-36) but not the other (vs 'Ear discharge symptoms'). The achievement of modest divergent validity under bi-factor also brings lowered convergent validity as might be expected (r at 0.251 for bi-factor 'Hearing' with binaural threshold), well below the declared cut-off 0.50. For 'Activities/healthcare' a similar competition or trade-off is seen between the two classes of validity, one which bi-factor solutions clarify. In summary, whilst some instances of divergent validity are also present and are slightly favoured by the best (bi-factor) solution

**Table 3. Pearson correlations (with 95% CIs) between external criterion variables and COMQ-12 total and specific factor scores.**

| External criterion variable → | Otomicroscopic findings | Weighted binaural aPTA | SF-36 1st PC total |
|---|---|---|---|
| **QUESTIONNAIRE VARIABLE, & Version of COMQ-12 score ↓** | **Active vs. Inactive** | **Transformed** | **Transformed** |
| **TOTAL 1st PC total** | 0.207 (0.084, 0.323) | 0.233 (0.111,0.348) | **0.547 (0.453, 0.629)** |
| **TOTAL Bi-factor general** | 0.167 (0.043, 0.287) | 0.247 (0.126, 0.361) | **0.566 (0.474, 0.645)** |
| **HEARING Simple CFA** | 0.177 (0.053, 0.295) | **0.315 (0.198, 0.424)** | 0.456 (0.350, 0.549) |
| **HEARING Bi-factor** | 0.112 (-0.013, 0.234) | **0.251 (0.130, 0.365)** | 0.043 (-0.082, 0.167) |
| **ACTIVITIES/ HEALTHCARE Simple CFA** | 0.122 (-0.003, 0.244) | 0.125 (0.0002, 0.247) | **0.419 (0.310, 0.517)** |
| **ACTIVITIES/HEALTHCARE Bi-factor** | 0.058 (-0.068, 0.181) | 0.023 (-0.102, 0.148) | **0.220 (0.098, 0.336)** |
| **EAR DISHARGE SYMPTOMS Simple CFA** | **0.357 (0.242, 0.461)** | 0.130 (0.005, 0.251) | 0.224 (0.102, 0.340) |
| **EAR DISCHARGE SYMPTOMS Bi-factor** | **0.335 (0.219, 0.442)** | 0.006 (-0.119, 0.131) | -0.014 (-0.139, 0.111) |

All variables use averages of Visit 1 and 2 data for reliability reasons (except for SF-36, see text and S1 Appendix). Correlations with the declared criterion measures (heads of columns) are emboldened. Weighted binaural auditory thresholds (air-conduction PTA) are the saved predicted values from the respectively most appropriate and fair binaurally weighted threshold models as specified in S3 Appendix. The SF-36 values denoting HRQoL have been inverted and transformed in normalising the SF-36 distribution (S4 Appendix). For bi-factor values only (so using the lower row for each field), six formal comparisons of correlation difference for documenting divergent validity were made between bold and non-bold entries using Fisher's Z in SPSS 26 [36]. Disregarding the totals and descending the two non-bold entries in each column in descending order of the other rows, these correlation magnitudes differed from the respective ((ie not predicted bi-factor) variables' emboldened values as follows. For Col 1: (Z = 2.400, p = 0.017; Z = 3.775, p<<0.001), for Col 2: (Z = 2.4444, p = 0.015; Z = 2.54, p = 0.011), and for Col 3: (Z = 1.894, p = 0.058; Z = 3.124, p = 0.002). We replicated this set of tests with the t-based Williams test [28], and only one difference in p greater than 0.0005 in the 4th decimal place of the p-value was obtained.

as expected, the divergence is patchy across measures and comes at the expense of convergent validity.

## Discussion

### Factor structure comparisons as pre-requisite to validation

Rigorous initial psychometric approaches to reliability, model fit and validity avoid wasted effort at the later stage of evaluation for clinical applicability, so do not conflict with diverse clinimetric ideals, even if delays for methodology frustrate clinicians [37]. From preceding studies and content validity considerations [10, 20], a solution with 3 relatively specific sub-scores was always promising for the 12-item set COMQ-12. However, the separate and important issue of what sub-scores really measure and whether 12 items could sample three obtained factors adequately for profiling had not previously been addressed. The best fit is produced by bi-factor solutions [31, 32] which suggests that sub-score profiling is not well supported in COMQ-12. On a small item set, bi-factor inevitably raises stability issues, and especially so when the item support for the structurally optimal number of factors becomes slight and not evenly spread. We showed this as a problem already with COMQ-12, even before the extra demands from the bi-factor solution.

The contrasting model classes raise the issue of what admixtures of a generic construct versus common method bias [38] are reflected in 1st PC versus the bi-factor solution's general factor. Here the only model with truly good fit was bi-factor, making it the default preferred structure for addressing both convergent and divergent validity. Bi-factor Root Mean Square Error Approximation (RMSEA) just failed to undercut the new excellence standard of ≤0.05 [33, 34], but the more stable Comparative Fit Index (CFI>0.950) met standard. The normalisation (in effect similar to Rsq) within the CFI makes CFI more stable than RMSEA across extreme conditions (such as having few fitted variables), hence perhaps more appropriate as a general guide. We confirmed expectation that bi-factor's better descriptive account would also

favour divergent validity by removing covariation with non-specific items referring to tinnitus and dizziness. The results for 'Hearing' indeed achieved this, but unfortunately the consequent reliability loss in the fewer items remaining with (high) loadings on the factors then compromised convergent validity. For both simple and bi-factor versions of a 'Hearing' sub-score in this data set, the validity correlations were low, even compared to the typically modest classical magnitudes of correlation found when audiometric hearing thresholds are used to predict reported hearing ability [39, 40]. For the 'Ear discharge symptoms' sub-score, the declared value for convergent criterion validity was also not met. Favourable recommendations for use at this point have to be restricted to the COMQ-12 total, and the sub-score 'Activities/health-care', which have shown acceptable convergent criterion validity and for the latter some divergent validity.

## General usefulness of bi-factor structures

The bi-factor general factor does not differ greatly from the 1st PC in this sample (r = 0.921), despite some items having low loading on it (six of twelve below 0.4, Table 2) so it can here be interpreted similarly as a weighted total. Bi-factor separation of generic components (including correlated response biases) from specific components offers widespread advantages in questionnaire measurement, and in health this distinction is natural and fundamental. Bi-factor solutions should also more flexibly serve a multiplicity of explicit measurement aims within the application-centred Clinimetric approach [37]. A recent well-executed Sino-Swiss study on a similar COM semi-generic QoL questionnaire used the bi-factor technique with a larger item set (M = 21) on an adequate sample size (N = 208) [22]. Those authors did not formally contrast solutions eg bi-factor against simple CFA, and only undertook preliminary forms of validation. However, their similarly good and interpretable bi-factor fit to ours, adds supports a bi-factor approach to PROMs in COM. Their not reporting some of the item sampling problems noted here probably reflects the model support of 4 specific (non-general) profiling factors from 21 items (average ratio 5.25), consistent with our suggestion that, together with the imbalance towards 'Hearing' in simple CFA, 3 factors from 12 (ratio 4.0) for COMQ-12 is too thin a spread to permit sub-score profiling.

## Neglect of divergent validity

For a total score, the lack of desired higher correlation with its criterion measure (than for the correlations with criterion measures adopted for sub- scores composing it) is inevitable and perhaps not too serious. This might explain the inattention in the literature to this more challenging form of validation, due to a wish to avoid discouragement through lack of divergence, in turn explaining the lack of development work on how divergent validity can be improved. Where the preliminary scientific goal is mere non-nullity, we see a major scientific threat of un-usable, uninterpretable, or ungeneralizable 'positive' findings, widely interpretable as publication bias [41]. The larger sample size needed for showing divergent validity via differences among modest correlations is one challenge. The pervasive item inter-correlation due to common method bias and response stereotypy in questionnaires is another, which without computer administration (ie making the items not co-visible) is hard to address. There is a pervasive bias towards high correlations when a single responding method is used [42], so high predicted (convergent) correlations may exaggerate the impression of a strong measure, unless a critical and analytical set of principles is applied, embracing bi-factor analyses and divergent validity. These principles seem not to have been heeded despite multi-disciplinary evidence: a decade ago, a similar example in orthopaedic PROMs reported low divergent validity of sub-scores from an otherwise 'promising' questionnaire [43].

## Strengths, limitations, and further research

The present work from the largest single-country COMQ-12 sample to date is the first to have explicitly and quantitatively compared 2-F with 3-F CFA, and simple with bi-factor CFA, also to address criterion validity. For both reliability and clinical relevance, we only proceeded to criterion validation after explicitly justified model choice. To avoid other obstacles to application, we also necessarily addressed linearity in two of the validity relationships, and handled skew in the same transformation, giving consequently more fair–but also stronger—assessment of correlation magnitudes. As a limitation, the under-sampling of content factors also limits the strength of methodological comparisons. We have avoided confirmation bias, critically and impartially examining a now widely translated and apparently used instrument. Given these rigorous provisions, some difficulties for 3-factor COMQ-12 sub-scores, particularly 'Hearing', cannot be brushed off.

We would encourage further criterion validation for COMQ-12, but the present evidence for 'Ear discharge symptoms' and 'Hearing' is strong enough to recommend first adding items to the former and restricting the latter to hearing disability items. The other items presently accommodated there (dizziness and tinnitus) could be represented either via bi-factor general or in some other homogeneous subset yet to be defined, but those make the present 'Hearing' sub-score from EFA or simple CFA uneasy mixture. For all PROM development, we would advocate explicitly reporting the trade-offs between interpretation, fit and parsimony of alternative models, and the use of demonstrably near-optimal models to assess the various forms of validity. Content validity form an interpretable factor solution is just a preliminary stage.

## Conclusions

1. The COMQ-12 item set in adult chronic otitis media is better modelled with 3 factors than 2 or 4; all 3-factor solutions examined gave reasonable interpretability, but only a bi-factor solution gave a high standard of fit.

2. The total COMQ-12 score showed acceptable convergent criterion validity for general clinical use as impact summary, but not sufficient divergent validity for research purposes. None of the 3 sub-scores achieved a satisfactory balance of convergent with divergent validity for either simple or bi-factor solution. Too limited item support, of which there was separate evidence, and non-specific response biases seem to be the main explanations.

3. Cautions over item support and other statistical constraints are needed by potential PROM users (eg about further forms of validity still needing to be demonstrated). Qualitative confirmation of overall content validity by an interpretable solution is a necessary start, but demonstration of convergent and divergent criterion validities is required for any sub-score profiling advocated.

## Supporting information

**S1 Appendix. Incorporation of replicate data from Phase 1 study and the use of dual-visit baseline.**
(DOCX)

**S2 Appendix. Scoring and handling of missing data.**
(DOCX)

**S3 Appendix. Definition of an appropriate binaural average hearing level (aPTA) for asymmetric hearing loss.**
(DOCX)

**S4 Appendix. Composition of total SF-36 score for present purposes.**
(DOCX)

**S5 Appendix. Minimum data set for present analysis.**
(SAV)

## Author Contributions

**Conceptualization:** Bojana Bukurov, Mark Haggard, Sandra Sipetic Grujicic.

**Data curation:** Bojana Bukurov, Nenad Arsovic, Sandra Sipetic Grujicic.

**Formal analysis:** Bojana Bukurov, Mark Haggard, Helen Spencer.

**Investigation:** Bojana Bukurov, Mark Haggard, Nenad Arsovic, Sandra Sipetic Grujicic.

**Methodology:** Mark Haggard, Sandra Sipetic Grujicic.

**Project administration:** Bojana Bukurov, Nenad Arsovic.

**Resources:** Bojana Bukurov, Helen Spencer, Nenad Arsovic.

**Software:** Mark Haggard, Helen Spencer.

**Supervision:** Mark Haggard, Helen Spencer, Nenad Arsovic, Sandra Sipetic Grujicic.

**Validation:** Mark Haggard.

**Visualization:** Helen Spencer.

**Writing – original draft:** Bojana Bukurov, Mark Haggard.

**Writing – review & editing:** Bojana Bukurov, Mark Haggard.

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
