## [Decision Letter · Decision Letter 0]

7 Jun 2022

PONE-D-22-07528Can short PROMs support valid factor-based sub-scores? Example of COMQ-12 in chronic otitis mediaPLOS ONE

Dear Dr. Bukurov,

Thank you for submitting your manuscript to PLOS ONE. After careful consideration, we feel that it has merit but does not fully meet PLOS ONE’s publication criteria as it currently stands. Therefore, we invite you to submit a revised version of the manuscript that addresses the points raised during the review process.

Please pay attention to the comments raised about statistical analysis and respond appropriately

We look forward to receiving your revised manuscript.

Kind regards,

Claudio Andaloro

Academic Editor

PLOS ONE

Journal Requirements:

Reviewers' comments:

Reviewer's Responses to Questions

**Comments to the Author**

1. Is the manuscript technically sound, and do the data support the conclusions?

Reviewer #1: Yes

Reviewer #2: Yes

Reviewer #3: Yes

2. Has the statistical analysis been performed appropriately and rigorously? 

Reviewer #1: I Don't Know

Reviewer #2: Yes

Reviewer #3: No

3. Have the authors made all data underlying the findings in their manuscript fully available?

Reviewer #1: No

Reviewer #2: Yes

Reviewer #3: No

4. Is the manuscript presented in an intelligible fashion and written in standard English?

Reviewer #1: Yes

Reviewer #2: Yes

Reviewer #3: Yes

5. Review Comments to the Author

Reviewer #1: Dear authors,

I would like to congratulate you for putting together such an interesting study. Validation of questionnaires, especially with regards for clinical relevance, are needed.

I have a few comments:

Introduction: "Nowadays, surgery is mostly elective in COM (to reduce hearing disability) and rarely obligatory (to prevent or arrest complications)"  I fully disagree with this statement. This has no scientific basis whatsoever. There is abundant evidence that intracranial complications of chronic otitis media, especially suppurative / cholesteatomatous are a significant cause of death. Lacks reference and a better explanation.

Introduction: "Balanced decision for surgery versus conservative management requires a multi-aspect appraisal to be communicated to the patient for shared understanding of realistic treatment gains in COM’s main impacts, and in generic HRQoL" - Again - although shared decision has a role in uncomplicated cases, that is not always the case. Again lacks references.

Methods, "participants and data acquired" - I assume these patients had unilateral COM, is that correct? Must be stated.

Reviewer #2: This is a well-written and technically sound paper that has explored the usefulness of patient-reported outcome measure through a questionnaire COMQ-12. The statistical methodology holds rigor and adds to its methodology.

Reviewer #3: This appeared to be a primarily straightforward manuscript (wrt. statistical analysis and data presentation). The material presented appeared insightful. I have some minor questions:

1. A sample size justification was provided, based on r = 0.40. However, (a) it is not clear whether it was based on the "primary response" variable (which should always be the case when presenting sample size/power statements, (b) the name of the statistical test was missing, (c) whether it was a 2-sided test that was used, and (d) no idea why an alpha = 0.001 was considered!

2. The statistical strategy section should also state what needs to be done when normality of model residuals were violated. Explain clearly, if continuous data, or discrete data is the focus. If continuous, assumptions of multivariate normality is immediate. Did the authors check whether that was satisfied, during actual data analysis?

3. Looks like data were generated for multiple time-points/repeated measures; so, why was some (repeated-measures) ANOVA-type approach not conducted, in addition? Some mixed-model approach would also have been appropriate; why was that not done? I was looking for some clarification.

6. PLOS authors have the option to publish the peer review history of their article (what does this mean?). If published, this will include your full peer review and any attached files.

Reviewer #1: No

Reviewer #2: No

Reviewer #3: No

---

## [Author Response · Author response to Decision Letter 0]

29 Jul 2022

Response to reviewers’ views

We thank the reviewers for their appreciation of the importance of the scientific issues addressed in our manuscript on COMQ-12, and for offering suggestions which definitely improve it. Below is our point-by-point response to each of the main comments. All changes that we have made from the original manuscript text are also given in the Tracked Changes (TC) version. Page- & line-pointers added here refer to the changes in the TCs- accepted version of the manuscript. Changes include usual general polishing for greater clarity and some cutting for wordcount. The latter stems partly from needing to make additions to address reviewers’ points, and this applies also to Supplementary information Appendix S2, the only Appendix in which the changes are substantial, but those are partly due also to the requirements of the Minimum Data Set. This minimum data set for present analysis according to PLOS instructions is uploaded now as Supplementary Information S5 Appendix and has an introduction in S2, naturally related to the issues explaining the general data structure and nature of certain variables.

There are no new introductions of substantive issues beyond those formerly present or raised by the reviewers. However, a single sentence clarifies the somewhat obvious, that the bi-factor general factor approximates simple 1st PC from EFA by giving their high correlation coefficient (> 0.90; P 12 line 297); this might perhaps be thought a substantive point as an r-value is a kind of result, and this was not mentioned before.

Reviewer Comments in Italic, Authors’ Responses in Roman font

Reviewer #1: Dear authors

I would like to congratulate you for putting together such an interesting study. Validation of questionnaires, especially with regards for clinical relevance, are needed.

I have a few comments:

Introduction: "Nowadays, surgery is mostly elective in COM (to reduce hearing disability) and rarely obligatory (to prevent or arrest complications)"  I fully disagree with this statement. This has no scientific basis whatsoever. There is abundant evidence that intracranial complications of chronic otitis media, especially suppurative / cholesteatomatous are a significant cause of death. Lacks reference and a better explanation.

We appreciate the endorsement of the need for proper validation. We had wanted to respect clinical relevance in the choice of criterion measures, and think that this was achieved, as the view seems to imply. 

We entirely agree that the quoted general sentence in the Introduction was unfortunately sweeping, and poorly worded in our revision of earlier draft, seeking brevity. It is readily improved by a clearer explanation and a reference. We had obviously not wanted to characterize all surgery, including for active disease, as elective. Of course, there is abundant evidence that both active forms of disease, mucosal and squamous, can lead to complications of COM (according to some series, up to 50%)*. But we here wished to convey a dimension of relative urgency of the recommendation for surgery and hope we have now achieved a good wording for that. In active COM, surgery is necessary, of course, but if not acute and progressing, it can be planned quasi-electively for a later date, and rarely is it urgent and obligatory (as, for example, it is with expected early development of extracranial or intracranial complications.) We are sorry for not seeing how our wording could have led to a misunderstanding. The rewritten sentence in the main text (Page 3 lines 51-56) tries to express this very succinctly, as it is a well-rehearsed issue. 

*Singh B, Maharaj TJ. Radical mastoidectomy: its place in otitic intracranial complications. J Laryngol Otol 1993; 107(12):1113–18.

Introduction: "Balanced decision for surgery versus conservative management requires a multi-aspect appraisal to be communicated to the patient for shared understanding of realistic treatment gains in COM’s main impacts, and in generic HRQoL" - Again - although shared decision has a role in uncomplicated cases that is not always the case. Again lacks references.

This is really a complementary aspect of the preceding point. We agree that shared decision for surgery is particularly appropriate in inactive forms of the disease. The sentence has now been modified and a relevant reference added; we think the intended dimension is more clearly conveyed now.

Methods, "participants and data acquired" - I assume these patients had unilateral COM, is that correct? Must be stated.

We appreciate that the reader might expect to see some mention of laterality within the general sample descriptions in Methods, so we have added that information on laterality was collected. The numbers of unilateral or bilateral disease of our patients are given in the Results section, in footnotes of Table 1. We had not dwelled on the categorical terms ‘unilateral/bi-lateral’, as this clinical shorthand description loses the more important information in the distributions of hearing levels (or other properties) on the two ears. 

Reviewer #2: This is a well-written and technically sound paper that has explored the usefulness of patient-reported outcome measure through a questionnaire COMQ-12. The statistical methodology holds rigor and adds to its methodology.

We appreciate this awareness of the careful reflection and effort that went into this work. We did our best to make a rigorous, balanced and not un-critical evaluation of the scientific worth and applicability of COMQ-12, emphasising the sometimes overlooked psychometric development issues. 

Reviewer #3: This appeared to be a primarily straightforward manuscript (wrt. statistical analysis and data presentation). The material presented appeared insightful. I have some minor questions:

1. A sample size justification was provided, based on r = 0.40. However, (a) it is not clear whether it was based on the "primary response" variable (which should always be the case when presenting sample size/power statements, (b) the name of the statistical test was missing, (c) whether it was a 2-sided test that was used, and (d) no idea why an alpha = 0.001 was considered!

We take the generally positive introduction from this reviewer in the spirit apparently intended, and we reciprocate. However, we cannot completely avoid commenting on some assumptions behind the points in the queries in explaining why we wrote as we did, or did not write, formerly. Some of these issues are not so small as the mere requests for more detail in reporting Methods imply. We first try to answer 1 (a)-(d) as briefly as simply and factually as possible, and later comment on how two, (a)-(d) are directed at other targets than those in the article.

(a) The type of power statement given is of generalised statistical power for an effect size (for which the correlation coefficient is the widely familiar example in an association). It is not an a priori sample size calculation grounded in a particular measure on a particular score. There is not space here to go into the mistaken view sometimes expressed that no other form of calculation than the latter should be done. In the past era of clinical trials, there was typically one dominating overall trial question (“is this treatment effective?”), for leading into policy decisions; this question was assumed to be most simply, if not entirely adequately, answerable by straightforward ‘significance’ of a single difference on a single measure. That principle became crystallised as a good-practice rule to help head off cherry-picking or confusion in reporting and applying trial results. The rule of a single predominant outcome measure (what we have taken ‘primary response’ to mean here) is in general long gone, but it remains true that clarifying relative importance of few study questions assists interpretation, as does pre-declaring the effect size magnitudes justifying conclusions. These things we have done. We had not intended to imply an a priori sample-size calculation, which is not called for in the study, as we clarify below, but to summarise and communicate a power scenario, and the modified wording emphasizes this more strongly. 

This study is not a clinical trial but a primarily correlational, psychometric study about a related set of measures from an item pool, where the concept of most important outcome measure does not have any obvious direct counterpart. No ‘response’ (ie mean pre-/post-treatment difference on an outcome measure after treatment) is reported here. The plurality of measures involved is so essential to the study question (supportability of multi-score profiling) that it figures in our title. We do, somewhat conventionally use the overall score (ie principal component, PC, as weighted total) as conceptual pivot and procedural step, we do conclude with recommending its use as being empirically supported for simple reasons of generality and reliability, and that recommendation might lead to use of COMQ-12 becoming concentrated only on this total. However, to have translated this generality into policy-relevant importance for a sample-size calculation based on a marginal p-value such as p=0.05, would have adversely restricted N for the whole set of issues in the psychometric study; it would have under-powered the study for the issue(s) actually at stake. The participant sampling adequacy is safe at N=246 and helps compensate for the higher variability (hence lower power) with shorter sub-scores. The explicit emphasis throughout is on the numbers of items per measure, which, counteracting some neglect in the PROMs literature, we here show is barely sufficient for the COMQ-12 under evaluation. This chief issue ‘Is there support-for-profiling with sub-scores?’ was prominently set out in the main text, but we have re-emphasised it in several places, chiefly in the strategy section P 5 lines 121-128 to head off any misunderstanding by readers.

(b) The simple statistical test for non-nullity of correlations (Fisher, a transformation of r to Z) is in widespread use and built into many statistical packages. We used the SPSS syntax, which is covered by the general SPSS declaration in the text; this declaration has become customary as default citation, and we did not consider it necessary to burden the bibliography with a reference to this almost universally used eponymous test. Nonetheless, we have now introduced small changes in wording to both text and table 3 footnotes to capture this. Extensions of Fisher’s Z are available for differences in correlation value, which is a less familiar analysis meriting more specification. We restrict that type of test to the quantification of divergent validity, largely covered in the footnote to Table 3, but have added there a background theoretical reference also for the issues in correlation differences.

(c) The short answer is: ‘necessarily, 2-tailed’. Even when some statistical authorities used to recommend conditions under which a single-sided test could be used, there were painful disputes over defining what those should be. The distinction introduces incentives to make results look more favorable than they were by producing diverse arguments for analyses meeting criteria for using 1-tailed tests. Many Bayesians regard the concept as meaningless, and that also has contributed to the use of 1-tail tests becoming less common than formerly. For these reasons, it seemed redundant to declare that all tests were 2-tailed, but we have now done this at our first given p-value, located at the power scenario referred to above. 

(d) There is a wealth of literature (suggested search-term ‘Replication Crisis’) from the last 15 years complaining much more vocally than in the previous 50 of inadequate statistical standards in biomedical and social science. This literature agrees that whatever the solution, p-values of 0.05 represent a generally low standard of evidence. In the light of this clean-up, any scientist is justified in aiming higher. There are circumstances where an a priori intention for alpha 0.001 could be appropriate, eg where high certainty is required. However, as implied under (a) above, our descriptive power scenario merely used this as one power parameter traded off against the other parameters in a scenario. We kept the values chosen within familiar ranges, so as to communicate the actually high power to readers who might not be thoroughly versed in the purpose and processes of power calculation. Throughout, we have allocated highest scientific priority to what should be considered useful effect magnitudes in respect of correlation coefficients, a question which is not N-dependent (apart from minor small-sample formula adjustments such as possible use of Williams t in preference to Fisher’s Z). Most importantly, as is present also in a priori sample size calculations, we made an explicit declaration of a set of value (s) pivotal for drawing conclusions. Those values have not been questioned by the reviewers and are transparent for debate. The power (ie against r=0) for the declared target absolute correlation values is extremely high and we do not see an alternative approach to such use of a set of calculation parameter values for communicating this fact which we felt it important for overall interpretation to report.

We furthermore in the final table report the procedures appropriate for statements about differences in correlation magnitude for divergent validity, although our main conclusions do not draw heavily on these. There is a prevalent bad habit (see for example Nieuwenhuis et al 2011) for authors to proceed, eg from presence versus absence of a ‘significant’ effect, to make difference-based or interaction statements without the appropriate statistical test, one for which their studies are mostly grossly underpowered. In the event, 5 of the 6 correlation differences in Table 3 are respectably significant and one marginal. The exact power for these differences is not a major issue. We had not declared that divergent validity, hence correlation differences, was our sole concern, but a supplementary consideration to convergent validity, so we do not deploy a second power scenario. More importantly, to head off cherry-picking of conclusions, we state that the magnitudes of the differences, although mostly conventionally significant by virtue of adequate sample size, are actually quite small, Accordingly, we conclude in Results and Conclusions only that some divergent validity is shown by the bi-factor solution, but modest and inevitably detracting from convergent validity as measured. The implication is that other instruments, and especially those constrained to be very brief for clinical reasons, need to be based on similarly rigorous and dispassionate validations. We imply that mostly they are not so, but the thrust of the article is not overtly or polemically critical. 

(2) The statistical strategy section should also state what needs to be done when normality of model residuals were violated. Explain clearly, if continuous data, or discrete data is the focus. If continuous, assumptions of multivariate normality is immediate. Did the authors check whether that was satisfied, during actual data analysis?

There is indeed some value in an explicit statement of measurement type, so we have simply inserted the words ‘for continuous measurement’ at one point. Our formerly omitting it is explained by its being the ‘un-marked’ common default instance in psychometrics, with categorical being the type requiring the explanation as inevitable etc; the Pearson correlations and conventional CFA require that measurement be continuous, and such omission where redundant is common in scientific writing. The account of item scaling to enhance the equal-interval properties at item level, referenced, and taking up much of Supplementary Information S2 Appendix, addresses the fullest justification of continuous measurement, and the declaration about that need not be labored in main text. With categorical measurement the concept would not arise.

Concerning multivariate normality (MVN), we had of course satisfied ourselves that the distributions were generally appropriate for the modelling. This was done in four now traditional ways to assure that distribution issues did not undermine any conclusions drawn: (a) graphic item raw distributions, (b) plotting residuals from component multiple regressions with single factor components of the overall model (which are at least true residuals), (c) habitually bootstrapping (BS), and also reporting the BS p-values for any normal-assumption p-values that are >0.01, or in appropriate instances for those that are in a magnitude range pre-defined as marginal; and (d) noting pairs of variables (items) with high absolute residuals, although this last is more about degrading potentially good fit (false-negative), rather than resisting spuriously good fit (false positive). We had not thought it appropriate to quote multivariate normality statistics, and the necessarily long following reasoning shows why not. Normality bears chiefly on the literal interpretation of p-values, something best played down anyway since the reforms of the 1980s and the 2010s. There are other important issues beyond apparent fit in model acceptance such as the df-ratio (which only the Akaike Information Criterion, by giving delta = (default minus saturated), well expresses – and we use that, but there was not space to emphasise specifically why; also, whether spending df on estimating intercepts and means was done to help handle missing data. We did not specifically report that decision or its reasoning, nor specify the– the ML estimator employed– again a somewhat default matter – etc etc. With so many detailed options, many not differing materially for a given set of data, it is not always clear where one should stop in reporting procedural detail and probably investigators with differing recent experiences would differ in their preferences. 

Nevertheless, we are very glad to have received this question on MVN as stimulus to formalize our approach. Immediately, we state the relevant action in the added paragraph to be found at the boundary between pages 9 and 10, lines 208-218. Some consequential minor refinements have contributed to the changes made to the Supplementary Information S2 Appendix. 

We think the reviewer is chiefly concerned about the explicit reporting of techniques used, so have tried to keep the following explanation brief. In short, simply stating whether MVN assumptions are ‘satisfied’ via cut-offs on two parameters (skewnesses, kurtoses – typically reduced to one p-value) could potentially be misleading by generalising false beliefs about the relevant data, these errors being of either false-positive or false-negative type. There is a need for systematic steps when multivariate normality is not safely met, but also if it is marginal. The steps should be ‘strategically’ guided and preferably pre-declared, but as they have also to respond to the obtained data, that ideal is not easily met, and their reporting does not sit easily under the introductory ‘strategy’ or other sub-section of Methods but will require explicit sections on preliminary analyses. This broader approach is required because of technical issues over MVN itself: the lack of consistent guidance on what value of Mardia index is to be taken as safe, marginal or dangerous; the software facilities using or giving access to raw variables rather than true residuals, and possible false assurance -- the fact that an overall index may conceal (‘false negative’) local violations of normality bearing upon the interpretation of local features.

In the light of the complexity of appropriate use of MVN information, when a colleague familiar with the package R-Lavaan returned to work recently, we asked her to implement our simple and bi-factor CFAs. This was as further check on our declared approach. To summarise: (a) the results are reproduced for both models on RMSEA and CFI indices to within 5 in the 3rd decimal place (that is, one half of one percent), and similarly on the p-values, particularly for the marginal links in the Hearing factor meriting this close attention; (b) Lavaan confirmed that the bi-factor model, whilst fitting much better, is also the more kurtotic; (c) in neither SPSS-AMOS nor R-Lavaan does the offered MVN index describe the residuals, but the raw variables; (d) in neither package is it easy to access the residuals for doing one’s own normality analyses, multi- or uni-variate. Shortly before our resubmission date she showed that residuals can be saved for further analysis and so accessed by the programming flexibility in R-Lavaan, but good guidance notes on the need and the process need to be provided. (e) On doing this, the MV skew became stronger than for raw data in the present dataset, but this is of less concern; the Mardia kurtosis that was highly significant in the raw values, though still positive (1.3) became NS (p=0.19) on the residuals. At time of resubmission these comparisons are ongoing with a view to the statistical applications note mentioned, so we have not felt it appropriate to modify the article to claim virtue of multi-versed checks by using more than one package. 

The challenge of this complexity and need for authors to nuance and devise an appropriate way to achieve economy in reporting is to produce just a few comprehensible sentences that are meaningful in the light of the complex reality. Our offering on pages 9-10 may still appear somewhat cryptic for the general reader, as much of the scientific reasoning behind what is appropriate has had for brevity to be excluded. However, we trust that it satisfies the reviewer’s emphasis on the need for comprehensive reporting; those of high statistical literacy might be in a position to judge appropriateness of our approach, ie concentrating various types of checking on where genuine issues of marginality for link retention might compromise conclusions. 

3. Looks like data were generated for multiple time-points/repeated measures; so, why was some (repeated-measures) ANOVA-type approach not conducted, in addition? Some mixed-model approach would also have been appropriate; why was that not done? I was looking for some clarification

The short answer is: not every aspect of a study’s data structure need appear in analyses in a single article and the article is already full. Indeed, it encourages general transparency to (briefly) declare other related data that are held but not analysed in the current paper, because not related to the scientific question, and this we did. We have now added a few further words of clarification to this effect. An article on the time/treatment aspect was submitted at around the same time to another journal and is now accepted there.

The reason we have not used repeated measures or other versions of ANOVA is that those techniques are for comparing mean effects (eg treatment) over time against background variability (individual and other) to assess the reliability and magnitude of change. The present article is primarily psychometric, so is not about such mean comparisons, shifts, effects etc, as in the later treatment aspect, but is about factor structure and appropriate scoring, using the more copious baseline data, and primarily the correlations between scores and available criterion measures. Later-visit data could at some complexity-cost also be used in development of measures (eg if manifest change led to concern that the measurement model should optimally span the periods from which measurements would be used); but, it is conventional to not do that, but rather to accept the structural account from baseline as defining the psychometrics. This is done because of sample attrition: first visit usually has the largest N of all the time-points and so best guarantees reliability, generality and the crucial stationarity of measurement, if perhaps not its optimality for capturing change. We share the enthusiasm for fitting categorical independent variables, where sampled, as random effects, eg by using mixed models; however the purpose and design here do not call for random-effects analyses either. 

We trust that these changes and the general clarifications of wording around them give the paper the appeal to the readership in the balanced way at which we must all aim.

---

## [Decision Letter · Decision Letter 1]

30 Aug 2022

Can short PROMs support valid factor-based sub-scores? Example of COMQ-12 in chronic otitis media

PONE-D-22-07528R1

Dear Dr. Bukurov,

We’re pleased to inform you that your manuscript has been judged scientifically suitable for publication and will be formally accepted for publication once it meets all outstanding technical requirements.

Kind regards,

Claudio Andaloro

Academic Editor

PLOS ONE

Additional Editor Comments (optional):

Reviewers' comments:

Reviewer's Responses to Questions

**Comments to the Author**

1. If the authors have adequately addressed your comments raised in a previous round of review and you feel that this manuscript is now acceptable for publication, you may indicate that here to bypass the “Comments to the Author” section, enter your conflict of interest statement in the “Confidential to Editor” section, and submit your "Accept" recommendation.

Reviewer #1: All comments have been addressed

Reviewer #2: All comments have been addressed

Reviewer #3: All comments have been addressed

2. Is the manuscript technically sound, and do the data support the conclusions?

Reviewer #1: Yes

Reviewer #2: Yes

Reviewer #3: (No Response)

3. Has the statistical analysis been performed appropriately and rigorously? 

Reviewer #1: Yes

Reviewer #2: Yes

Reviewer #3: (No Response)

4. Have the authors made all data underlying the findings in their manuscript fully available?

Reviewer #1: Yes

Reviewer #2: Yes

Reviewer #3: (No Response)

5. Is the manuscript presented in an intelligible fashion and written in standard English?

Reviewer #1: Yes

Reviewer #2: Yes

Reviewer #3: (No Response)

6. Review Comments to the Author

Reviewer #1: Thanks for the clarifications. I have no further concerns. Congratulations for the authors for the manuscript.

Reviewer #2: The revisions carried out are satisfactory and have suitably addressed all the queries raised during the review process.

Reviewer #3: (No Response)

7. PLOS authors have the option to publish the peer review history of their article (what does this mean?). If published, this will include your full peer review and any attached files.

Reviewer #1: **Yes: **Rafael da Costa Monsanto

Reviewer #2: **Yes: **Mainak Dutta

Reviewer #3: No

---

## [Editor Report · Acceptance letter]

20 Sep 2022

PONE-D-22-07528R1 

Can short PROMs support valid factor-based sub-scores? Example of COMQ-12 in chronic otitis media 

Dear Dr. Bukurov:

I'm pleased to inform you that your manuscript has been deemed suitable for publication in PLOS ONE. Congratulations! Your manuscript is now with our production department. 

Kind regards, 

on behalf of

Dr. Claudio Andaloro 

Academic Editor

PLOS ONE